# Cross-Modal Feature Learning for Point Cloud Classification

1st Wan-Hua Li
*the College of Computer and Data Science*
*Fuzhou University*
Fuzhou, China
liwanhuazi@163.com

2nd Chun-Yang Zhang
*the College of Computer and Data Science*
*Fuzhou University*
Fuzhou, China
zhangcy@fzu.edu.cn

*Abstract*—Traditional 3D shape classification methods face challenges due to the complexity and variability of point cloud data. To address this issue, we propose CFL framework that integrates proxy weights from two modalities through an average fusion approach and adopts a proxy-based contrastive learning strategy to enhance feature representation. By using the average fusion method. we can effectively capture both texture features and geometric features via integrating complementary information from different modalities. Furthermore, the proxy-based contrastive learning method is designed to acquire representations by learning a unified space. Experimental results demonstrate that our CFL method significantly improves classification performance on the ModelNet10 dataset. Meanwhile, we conduct ablation studies on ModelNet10 dataset to confirm the pivotal role of the average fusion method and the proxy-based contrastive learning method, highlighting the potential of cross-modal feature learning method in advancing 3D shape classification tasks.

*Index Terms*—3D shape classification, geometric complexity, cross-modal feature learning, proxy weight

## I. INTRODUCTION

While 2D classification has seen substantial advancements [1] [2] with Deep Neural Networks (DNNs), 3D deep learning techniques [3] are still performing poorly on 3D classification tasks, which mainly caused by the unique geometric complexities of 3D point cloud data. This complexity poses a challenge in preserving the intrinsic properties of 3D shapes during the learning process, making it more difficult for traditional models to achieve high accuracy.

Existing 3D point cloud classification methods typically rely only on the inherent geometric information of the point cloud, ignoring the valuable color and texture cues that can be gleaned from associated images. Such problem can result in a less comprehensive representation of the visual environment due to a lack of multimodal integration, which is particularly pronounced in applications that require broad classification capabilities, such as autonomous navigation [6] or augmented reality [7], where our research aims to explore. Therefore, it's urgent to find an robust model that can effectively integrate cross-modal information, improving its adaptability under different circumstances.

In this paper, we propose a Cross-Modal Feature Learning(CFL) framework to tackle the shortage of existing 3D point cloud classification methods. We first feed 3D point cloud data into Dynamic Gragh CNN(DGCNN) [8], which

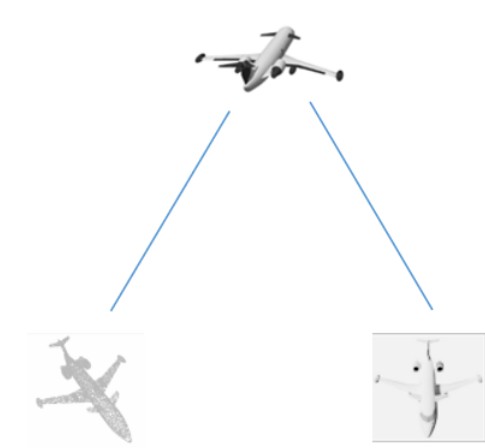

Fig. 1: Comprehensive visualization of the 3D object and rendered image within our framework.

outputs features and proxy weights of the traing point cloud data. For the corresponding rendered images, we employ a pretrained CLIP [9] model, leveraging its image encoder to extract the features and proxy weights of images. Specifically, our CFL framework leverages the complementary strengths of both 3D point clouds and 2D images by incorporating their features via an average fusion method [10]. This approach allows for a dynamic and context-aware interaction between the spatial and visual modalities, facilitating a deeper understanding of the scene's geometric and textural properties. Moreover, we employ a proxy-based contrastive learning method to align samples with proxies, which can enhance the model's ability to distinguish between different classes by emphasizing the differences between class-level representations.

Experimental results illustrate that our proposed method significantly improves point cloud classification performance compared to existing approaches [11] [12]. Specifically, our model demonstrates superior accuracy, outperforming several state-of-the-art methods in 3D point cloud classification tasks. Such improvement is particularly notable when compared to traditional techniques, highlighting the effectiveness of our approach. Furthermore, ablation studies reveal the critical roles

of the average fusion mechanism and proxy-based learning method in enhancing feature representation. These studies confirm that the integration of proxy weights through average fusion allows our model to capture both geometric and texture features more effectively, while proxy-based contrastive learning method further refines the feature representation by learning a unified space. Our CFL framework not only boosts classification performance but also provides deeper insights into the importance of these methods in achieving advanced 3D shape classification.

Our CFL framework has three major advantages:

- We propose a novel framework that effectively integrates point clouds and images using an average fusion method. This integration allows our model to leverage complementary information from both point clouds and images, enhancing the feature representation and improving classification performance.
- Our CFL framework employs a proxy-based learning method that aligns and enhances fused features in point clouds and images. This technology improves the discriminative ability of learned features, thereby significantly improving the performance of point cloud classification.
- Our approach outperforms multiple baseline methods. Experimental results demonstrate the effectiveness and robustness of our proposed framework.

## II. RELATED WORKS

In this section, we provide an overview of the existing works related to the study of our proposed framework. We begin by examining the current state of the art in 3D point cloud classification, a field that has seen significant advancements and serves as a critical component in our approach. We then delve into the cross-modal representation learning, which is essential for understanding how features from different modalities can be effectively integrated to enhance model performance.Finally, we explore contrastive learning, a powerful technique that has gained considerable attention for its ability to learn robust feature representations.

### A. 3D Point Cloud Classification

When it comes to 3D point cloud classification, previous studies typically fallen into two main types: 1) projection-based methods and 2) point cloud-based methods. Projection-based methods [13], [14] convert 3D point clouds into 2D representations such as range images or depth maps. These methods usually rely on multi-view representations of point clouds, which are very beneficial for capturing view-dependent features of the data.In contrast, point cloud-based methods [15] [8] focus directly on raw point cloud data, exploiting the inherent spatial relationships and local neighborhood structures of point clouds. These methods have evolved significantly with the advent of deep learning, especially with the development of point-based neural networks such as PointNet [3], Point-Net++ [15] and DGCNN. Such networks are able to capture global features from point cloud data without relying on any predefined mesh structure.

### B. Cross-Modal Representation Learning

In the realm of cross-modal representation learning, previous studies have explored various techniques to bridge the gap between different types of data, such as text, images, and 3D point clouds. These methods aim to learn a unified representation that captures the essence of each modality and facilitates the transfer of knowledge across domains. For example, some methods [16] [17] combine 3D point cloud data with other modal data, leveraging the spatial and geometric information of point clouds to enhance the representation capabilities. These approaches not only utilize the strengths of individual modalities but also address the challenges of modality-specific features that may not be fully captured when using a single type of data. Notable approaches include fusion strategies that combine point cloud features with image-based features using neural networks, such as the ULIP [19] which has shown superior performance in cross-modal applications by effectively combining complementary properties of different data sources. This fusion enables a richer and more robust representation, facilitating improved generalization and performance in tasks that require understanding and integrating multiple forms of data. Such advancements underscore the importance of cross-modal techniques in pushing the boundaries of representation learning across diverse and complex datasets.

### C. Contrastive learning

Contrastive learning has become a foundational approach in representation learning, particularly for its ability to learn robust embeddings by distinguishing between positive and negative data pairs. Traditional methods like SimCLR [20] and MoCo [21] have demonstrated success in visual domains by maximizing the agreement between positive pairs while minimizing it between negative pairs, which often requires large batch sizes and complex sampling strategies. Contrastive learning has been extended to other domains, including 3D point clouds and cross-modal data. PointContrast [22] has been designed to handle the unique challenges of point cloud data by leveraging its spatial structure to learn invariant representations. However, traditional contrastive learning methods only consider the distance of sample-to-sample. To address this problem, Proxy-based Contrastive Learning method has been introduced, which compare samples to proxies, thus maintaining performance. In our CFL framework, we employ this proxy-based approach to efficiently integrate and enhance features from different modalities, ensuring the learned representations are both robust and discriminative.

## III. PROPOSED METHOD

In this section, we first detail the methods used for feature extraction from point cloud data and images, which are essential for the subsequent steps of our proposed method. Thereafter, we explore the intricacies of the average fusion mechanism. This mechanism is crucial in our approach as

it promotes interactions between different modalities, thereby enhancing the learning process. Subsequently, we turn our attention to the proxy-based learning approach, which is another cornerstone of our research. Fig. 2 illustrates the comprehensive overview of the CFL framework.

## A. Class Distribution Alignment

We introduce a novel classification loss function designed to mitigate the impact of class imbalance introduced by FocalLoss [24].Our approach to classification is fundamentally anchored in directing increased attention towards underrepresented categories with fewer sample instances. Borrowing insights from existing scholarly work, we have embraced the mathematical framework from DLSA [25] and FocalLoss to rebalance the class distribution. This is achieved through a sophisticated mechanism that recalibrates the class weights, detailed as follows:

$$\omega(i) = \frac{n_i^{-q}}{\sum_{j=1}^{K} n_j^{-q}}, \tag{1}$$

where $n_i$ represents the number of samples in the $i$-th category, and $q$ is a positive parameter that regulates the influence of sample counts on the weight distribution. To address the challenge of class imbalance, we integrate the class-specific weighting with the initial classification loss function, as detailed in Eq.2. This refined approach allows for a more nuanced treatment of the imbalance, ensuring that each class contributes appropriately to the learning process. The revised weighted classification loss, which takes into account this class-wise weighting, is articulated in the following manner:

$$\mathcal{L}_{CLS}(\mathcal{B}) = -\sum_{x \in \mathcal{B}} \omega(y) \log L(\theta; x), \tag{2}$$

where $\mathcal{B}$ is the set of samples within a batch. This formulation ensures that the classification loss is adjusted in accordance with the prevalence of each class, thereby providing a more balanced optimization objective that mitigates the impact of class imbalance.

## B. Feature and Proxy Weight Extraction

For point clouds, we employ DGCNN [8] as the encoder, which is good at capturing local and global features of unstructured point cloud data. DGCNN processes point cloud data through dynamic graph convolution to provide rich feature representation and proxy weight for subsequent modal fusion. Meanwhile, we leverage the image encoder from CLIP, which has been pre-trained on a large number of image-text pairs, providing powerful visual feature representations that significantly support our cross-modal learning efforts.Therefore, we can capture the features of point cloud $P_i$ and corresponding rendered image $I_i$ as follows:

$$f_i^P = f_P(P_i), \tag{3}$$

$$f_i^I = f_I(I_i), \tag{4}$$

where $f_P$ is the feature extractor for the point cloud and $f_I$ is the frozen CLIP extractor for the image. Moreover, we extract

TABLE I: Experimental results on different contrastive learning methods.

| Method | ModelNet10 | Shapenet10 [23] | Avg. |
|--------|------------|-----------------|------|
| PCL | **97.8** | **45.6** | **71.7** |
| CL | 92.6 | 45.5 | 69.1 |

point cloud proxy weight $w_P$ as well as image proxy weight $w_I$. These proxy weights serve a dual purpose: they not only emphasize the most informative features in each modality, but also implement a dynamic weighting mechanism that enhances the learning process. By leveraging the DGCNN encoder, we exploit its ability to capture fine-grained geometric details and spatial hierarchy of point clouds to generate proxy weights that reflect the importance of each point in the cloud. Meanwhile, the CLIP encoder, which is extensively pre-trained on a diverse image dataset, generates proxy weights that encapsulate both the visual saliency and semantic richness of an image.

## C. Average Fusion Mechanism

In our framework, the average fusion mechanism plays a key role in integrating the features of point clouds and images. This mechanism involves computing the average of the agent weights obtained from the two modalities to create a fused super agent.The integration of these proxy weights allows our CFL framework to adaptively focus on the most relevant features during training, which is particularly beneficial for handling the variability and geometric complexity present in real-word. The average proxy weight is illustrated as follows:

$$w_{Avg} = (w_P + w_I)/2 \tag{5}$$

By averaging these weights, we create a fused super agent $w_{Avg}$ that effectively combines the complementary information from both modalities. This integration enables our CFL framework to simultaneously capture the rich texture features inherent in image data and the detailed geometric features characteristic of point cloud data. Moreover, the adaptive nature of the average fusion method allows our framework to dynamically adjust its focus on the most relevant features throughout the training process. This adaptability is crucial for managing the inherent variability and complex geometric structures found in real-world 3D datasets. By emphasizing the most pertinent features, the average fusion method enhances the robustness and generalization capabilities of our model, ensuring that it performs well across different scenarios and datasets. This capability to adapt and integrate diverse types of information is a key factor in the superior performance of our CFL framework in 3D shape classification tasks.

## D. Proxy-based Contrastive Learning

When directly applying traditional contrastive-based learning methods to poing cloud classification, we observe a significant drop in performance, as evidenced by the results in Table I. The main factor is that most contrastive-based learning

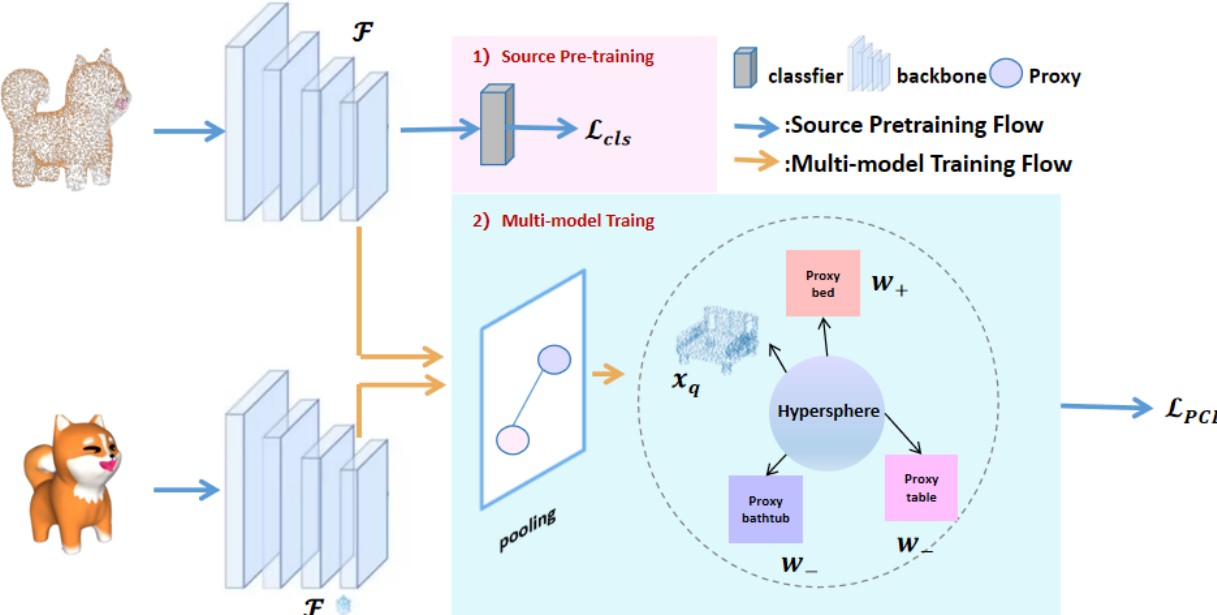

Fig. 2: The overall view of the Cross-modal Feature Learning (CFL) framework. The framework integrates proxy weights from two distinct modalities—point cloud and image data—using an average fusion approach. This approach effectively amalgamates complementary information, capturing a rich set of texture and geometric features. The framework is further enhanced by a proxy-based contrastive learning strategy, which operates within a unified feature space to refine the representational capabilities of the model. The harmonious integration of these components facilitates the learning of a comprehensive feature space that underpins the superior classification performance of our method on the ModelNet10 dataset.

methods focus only on sample-to-sample pairs, which can fail to capture the intricate relationships and spatial distributions within point clouds. To tackle this issue, we propose a proxy-based learning method specifically designed for 3D point cloud classification, which is inspired by the principles of PCL [26]. In this method, proxies serve as stable anchors, enabling the model to learn more robust and representative features by aligning samples with these proxies. We first project both sample features $f_i^P$ and proxy weights $w_{Avg}$ onto a projection layer, defined as follows:

$$e_i = h\left(\boldsymbol{f}_i^P\right), \tag{6}$$

$$v_i = g\left(\boldsymbol{w}_{Avg}\right), \tag{7}$$

where h($\cdot$) is a three-layer MLP used for sample embedding projection, and g($\cdot$) is a single-layer MLP used for proxy weight projection. The core idea of our approach is to minimize the distance between the proxy and the positive sample. The proxy-based contrastive learning loss is defined as:

$$\mathcal{L}_{\text{PCL}} = -\frac{1}{N}\sum_{i=1}^{N}\log\frac{\exp\left(\boldsymbol{v}_c^{\top}\boldsymbol{e}_i\right)}{E}, \tag{8}$$

where $E$ is represented as follows:

$$E = exp(v_c^T e_i) + \sum_{i=1}^{C-1} exp(v_k^T e_j) + \sum_{j=1,j\neq i}^{B} exp(e_i^T e_j). \tag{9}$$

### E. Overall loss

The overall loss function of our CFL framework is a combination of classification loss and proxy-based learning loss, which can not only accurately classify but also learn effective feature representations. The overall loss is defined as:

$$\mathcal{L}_{total} = \mu\mathcal{L}_{CLS} + \lambda\mathcal{L}_{PCL}, \tag{10}$$

where $\mu$ and $\lambda$ are the weights assigned to the classification loss $\mathcal{L}_{CLS}$ and proxy-based learning loss, respectively.

## IV. EXPERIMENTS

In this section, we evaluate the performance of our proposed method on the ModelNet10 [27] dataset, a widely recognized benchmark for 3D object classification. We begin by offering a comprehensive description of the ModelNet10 dataset, highlighting its selection for our experiments. Following this, we delve into the specifics of our implementation. We then present the main results of our experiments, showcasing the efficacy of our approach through a meticulous comparison with state-of-the-art techniques and baseline models. We conduct a thorough ablation study to dissect the contributions of various components of our method, gaining insight into their respective impacts on the overall performance.

### A. Datasets

ModelNet10 is a widely recognized benchmark dataset for 3D shape classification, comprising 5039 3D CAD models

TABLE II: Results on ModelNet10 under the CFL settings

| Rep. | Method | ModelNet10 |
|---|---|---|
| 2D projection | PANORAMA-ENN [28] | 96.85 |
| Point Cloud | SO-Net(CVPR'18) [11] | 95.50 |
| | KCNet [12] | 94.40 |
| | PCNN [29] | 94.40 |
| | Cross-atlas [30] | 91.20 |
| | PolyNet [31] | 91.20 |
| | SUG [32] | 96.7 |
| Ours | CFL | **97.8** |

from 10 diverse categories, including common household items such as beds, chairs, desks, and sofas. The dataset is meticulously partitioned into 4,183 training samples and 856 testing samples, providing a robust framework for evaluating the classification accuracy of algorithms. Furthermore, as shown in Fig. 1, we generate a set of 12 rendered images per point cloud, capturing the 3D object from various viewpoints and enhancing the model's ability to generalize across different perspectives. This comprehensive approach not only improves the dataset's utility for training but also challenges the model to learn from multiple aspects of the object, thereby improving its discriminatory capabilities. The strategic combination of point cloud data and rendered images, alongside our implementation of the CFL method, has proven to be a winning formula for achieving exceptional classification results on the ModelNet10 dataset.

*B. Implementation Details*

For the point cloud data, we randomly sample 2048 points from each 3D model and normalize them to fit within a unit sphere. In addition, we render multiple 2D views of each 3D model and resize the images to 224x224 pixels. For the optimization of our model, we select the Adam optimizer, a widely recognized algorithm that adapts the learning rate based on the average of recent gradients. We initialize the learning rate at 0.001, a value that has been empirically shown to provide a robust starting point for a wide range of tasks. Additionally, we employ weight decay with a coefficient of 0.00005 to regularize the model and mitigate the risk of overfitting. Furthermore, we use a batch size of 32 and train the model for 400 epochs.Furthermore, we maintain a detailed log of training metrics, including loss values, accuracy, and other relevant indicators, to monitor the model's progression and to identify any potential issues during training. This comprehensive approach to the training process is fundamental to the success of our model in classifying the diverse categories present within the ModelNet10 dataset.

*C. Main Results*

We evaluate our proposed CFL framework on the Model-Net10 dataset, comparing its performance with several methods across both 2D projection and point cloud categories. Ta-

ble II summarizes the classification accuracy of these methods, highlighting the effectiveness of our approach.

In the 2D projection category, the PANORAMA-ENN method achieves an impressive accuracy of 96.85%. The proposed PANORAMA-ENN method fully exploits the power of multi-view representation to convert 3D point clouds into 2D images, thereby effectively capturing view-dependent features. While this approach is beneficial for extracting texture and shape cues from different angles, it may fall short in fully leveraging the spatial structure inherent in raw 3D point cloud data. For point cloud-based methods, several advanced techniques are evaluated. For example, SO-Net performs an accuracy of 95.50%. This method utilizes a self-organizing map (SOM) to learn hierarchical spatial structures within point clouds, providing a robust representation of the data. KCNet and PCNN, both of which focus on point-wise feature learning, reported accuracies of 94.40%. These methods demonstrate the challenges of directly working with point cloud data, where capturing local geometric details and global context simultaneously can be difficult. Despite Cross-atlas and PolyNet's innovative approaches to processing point cloud data, they achieve a relatively low accuracy of 91.20%, suggesting potential limitations in their ability to generalize across a wide range of 3D shapes, especially when dealing with complex and irregular geometries.The SUG method, which introduces a novel approach to utilizing geometric features, performed notably better with an accuracy of 96.7%, underscoring its strength in capturing fine-grained geometric details in point clouds.

Our CFL approach, as shown in the last row of Table II, achieves a top accuracy of 97.8% on the ModelNet10 dataset, surpassing all the compared methods. This significant improvement can be attributed to the unique strengths of our framework. Specifically, the integration of cross-modal feature learning in our CFL framework allows for the effective fusion of complementary information from 2D images and 3D point clouds. By doing so, our model can capture both texture and geometric features, leading to richer and more discriminative representations. This cross-modal synergy is particularly powerful in addressing the geometric complexity and variability inherent in 3D shapes. Moreover, class distribution alignment method proposed in our method addresses the issue of class imbalance, which is often a challenge in 3D shape classification tasks. By assigning appropriate weights to different classes, our model is able to focus more on underrepresented categories, ensuring a more balanced and accurate classification across all classes. Finally, we use a proxy-based contrastive learning strategy to enhance the feature representation by efficiently distinguishing between different classes. This approach reduces computational complexity while maintaining high performance, ensuring that the learned representations are both robust and discriminative.

Overall, the experimental results clearly demonstrate that our CFL framework not only achieves superior performance on the ModelNet10 dataset but also sets a new benchmark for 3D shape classification.

TABLE III: Class-wise accuracy studies on ModelNet10

| Method | PCL | Image | Bathtub | Bed | Bookshelf | Cabinet | Chair | Lamp | Monitor | Plant | Sofa | Table | Avg. |
|--------|-----|-------|---------|-----|-----------|---------|-------|------|---------|-------|------|-------|------|
| CFL | | | 86.7 | 100 | 100 | 91.5 | 98.7 | 92.8 | 96.7 | 92.5 | 96.0 | 97.9 | 95.3 |
| | ✓ | | 76.4 | 100 | 97.4 | 86.2 | 96.7 | 96.7 | 97.8 | 86.9 | 97.3 | 99.0 | 93.4 |
| | ✓ | ✓ | **87.9** | 98.7 | 99.0 | 84.3 | **100** | **100** | 97.6 | **99.4** | 95.7 | **100** | **96.3** |

## D. Ablation Studies

To further investigate the effectiveness of our proposed CFL framework, we conduct a series of ablation studies on the ModelNet10 dataset, focusing on the class accuracy of different model configurations. Table III shows the results of these experiments, which aim to isolate the impact of key components on the overall performance of the model.

The first row in Table III reports the class-wise accuracy of SUG which plays a role of baseline in our CFL framework. In this scenario, the model achieved an average accuracy of 95.3%, with certain classes like Bed and Bookshelf reaching perfect accuracy of 100%. However, performance on more geometrically complex classes such as Bathtub and Cabinet was relatively lower, indicating the limitations of relying solely on point cloud data for 3D shape classification. Specifically, when we add the proxy-based contrastive learning method to our framework, the average class-wise accuracy reaches a superior result of 93.4%. However, although the configuration demonstrates high performance in classes like Chair and Sofa , it struggled with the Bathtub class, achieving only 76.4% accuracy. This suggests that while proxy-based contrastive learning method is effective for capturing the spatial structure of certain objects, it may not fully exploit the textural information that is beneficial for differentiating between geometrically similar classes. The last row in Table III demonstrates the results when both point cloud and image modalities are integrated within the CFL framework. This configuration yielded the highest average accuracy of 96.3%, with several classes achieving perfect accuracy. The accuracy improvements across most classes, particularly the challenging Bathtub and Plant classes, underscore the efficacy of our cross-modal fusion strategy. This integration effectively leverages the complementary strengths of both modalities, leading to more robust and discriminative representations.

These ablation studies highlight the significant contributions of each component within the CFL framework. The results clearly demonstrate that the combination of point cloud and image data within a unified cross-modal learning framework provides a substantial performance boost, enabling the model to capture both geometric and textural nuances across different 3D object classes.

## V. CONCLUSIONS AND FUTURE WORK

In this paper, we propose a new point cloud classification method that integrates point clouds and images by using average fusion method and proxy-based learning techniques. Our method exploits the complementary information of the two modalities to enhance feature representation and classification accuracy. We conduct experiments on the ModelNet10 dataset, a widely used benchmark for 3D shape classification. Moreover, we compare our results with several baseline methods and show significant performance improvements. Ablation studies further reveal the importance of average fusion mechanisms and proxy-based contrastive learning in optimizing feature integration and discrimination. These findings highlight the effectiveness of our approach and its potential to advance multimodal learning in the field of 3D shape recognition.

In the future, we aim to extend our research in several promising directions. For instance, we plan to explore more advanced fusion methods that can capture the intricate relationships between point clouds and images, potentially leading to more robust and discriminative feature representations. Moreover, we are interested in applying our classification method to more complex and diverse 3D datasets, such as ShapeNet. This will not only test the generalizability of our approach but also allow us to evaluate its performance under various conditions.

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
