# OpenReview forum: "Cross-Modal Feature Learning for Point Cloud Classification"
_IEEE.org/ICIST/2024/Conference — IEEE ICIST 2024 Conference Submission_

### Official Review · Reviewer_6yxX · 2024-08-21
**Accept**

**Rating:** 7
**Confidence:** 5

**Review:**

The paper presents a novel CFL (Cross-modal Fusion and Learning) framework that aims to tackle the challenges in 3D shape classification arising from the complexity and variability of point cloud data. By integrating proxy weights from multiple modalities through average fusion and adopting a proxy-based contrastive learning strategy, the proposed method enhances feature representation and achieves improved classification performance. There are some suggestions:
1.The authors should compare their method with more recent and state-of-the-art 3D shape classification approaches to position their work within the broader research landscape.
2.It would be interesting to discuss limitations and potential directions for future work.

---

### Official Review · Reviewer_rwBy · 2024-08-24
**This article is very interesting and a good one.**

**Rating:** 7
**Confidence:** 5

**Review:**

In this paper,  a new point cloud classification method that integrates point clouds and images by using average fusion method and proxy-based learning techniques is proposed. The obtained result is valuable and can be accepted if the following problems can be clarified.
1. What's the difference between the 'Introduction' and the 'Related Works' ?
2. The motivations should be further highlighted in the manuscript, e.g., what problems did the previous works exist? How to solve these problems? The authors may consider analyzing the problems of the previous works and how to address these problems with the proposed method.
3. The format of references needs to be uniform.
4. The quality of language needs significant improvement, and professional editing may be necessary.

---

### Official Review · Reviewer_F5fk · 2024-08-25
**This work provides new insight into the development of advanced multimodal fusion techniques for improving 3D point cloud classification through integrated point cloud and image data**

**Rating:** 7
**Confidence:** 4

**Review:**

In the manuscript titled "Cross-Modal Feature Learning for Point Cloud Classification"introduces the CFL framework, which enhances 3D shape classification by integrating texture and geometric features through average fusion and proxy-based contrastive learning, achieving significant performance improvements. This work provides new insight into the development of advanced multimodal fusion techniques for improving 3D point cloud classification through integrated point cloud and image data.However, lack of sufficient date to  prove the rationality of the conclusion .It is recommended to add sufficient dates,further justify the proposed method.

---

### Decision · Program_Chairs · 2024-09-08

Accept (Oral)